# Extending the SBUV PMC Data Record with OMPS NP

Matthew T. DeLand[1] and Gary E. Thomas[2]

[1]Science Systems and Applications, Inc. (SSAI), Lanham, Maryland 20706  USA

[2]Laboratory for Atmospheric and Space Physics (LASP)/University of Colorado, Boulder, Colorado 80303  USA

Correspondence to:  Matthew DeLand (matthew.deland@ssaihq.com)

**Abstract.**  We have utilized Solar Backscatter Ultraviolet (SBUV) instrument measurements of atmospheric radiance to create a 40-year record of polar mesospheric cloud (PMC) behavior. While this series of measurements is nearing its end, we show in this paper that Ozone Mapping and Profiling Suite (OMPS) Nadir Profiler (NP) instruments can be added to the merged SBUV PMC data record.  Regression analysis of this extended record shows smaller trends in PMC ice water content (IWC) since approximately 1998, consistent with previous work.  Current trends are significant at the 95% confidence level in the Northern Hemisphere, but not in the Southern Hemisphere.  The PMC IWC response to solar activity has decreased in the Northern Hemisphere since 1998, but has apparently increased in the Southern Hemisphere.

## 1. Introduction.

Determination of long-term (multi-decadal) variations in the Earth's mesosphere (60-100 km) is challenging.  In situ measurements can only be made by rockets that provide a brief snapshot of local conditions.  Ground-based measurements of key parameters (e.g. temperature, water vapor, winds) are only available at selected locations.  While some data sets are quite long (e.g. phase height (Peters et al. (2017)), other potentially valuable data sets have gaps.  Some relevant satellite datasets do exist (e.g. Upper Atmospheric Research Satellite (UARS) Halogen Occultation Experiment (HALOE) (Hervig and Siskind, 2006), Aura Microwave Limb Sounder (MLS) (Lambert et al., 2007; Schwarz et al., 2008), and Thermosphere-Ionosphere-Mesosphere Energetics and Dynamics (TIMED) Sounding of the Atmosphere using Broadband Radiometry (SABER) (Remsberg et al., 2008).  However, since the lifetime of a single instrument is

generally limited to 10-15 years, maintaining continuity for a specific parameter over multiple
decades again becomes an issue.

Another option is to measure an observable quantity that provides indirect information about the
background state of the mesosphere. Polar mesospheric clouds (PMCs) are observed only at
high latitudes (typically >50°) and high altitudes (80-85 km) during summer months in each
hemisphere. They are formed from small ice crystals (~20-80 nm radius), whose formation and
evolution are very sensitive to the temperature (< 150 K) and water vapor abundance near the
mesopause. Recent work (e.g. Hervig et al. (2009), Rong et al. (2014), Hervig et al. (2015),
Berger and Lübken (2015), Hervig et al. (2016)) has shown quantitative relationships between
PMC observables (occurrence frequency, albedo, ice water content) and mesospheric
temperature and water vapor.

The Solar Backscatter Ultraviolet (SBUV) instrument (Heath et al., 1975) was originally
launched on the Nimbus-7 satellite in 1978 to measure stratospheric profile and total column
ozone, using nadir measurements of backscattered UV radiation between 250-340 nm at
moderate spatial resolution (170 km x 170 km footprint). Thomas et al. (1991) showed that these
measurements could also be analyzed to identify bright PMCs as an excess radiance signal above
the Rayleigh-scattered sky background, modified by ozone absorption. These measurements
have been extended by the second generation SBUV/2 instrument, which has been flown
successfully on seven NOAA satellites from 1985 to the present. DeLand et al. (2003) describes
the extension of the SBUV PMC detection algorithm to SBUV/2 measurements. We use the
general term "SBUV" to describe these instruments unless a specific satellite is being discussed.
All SBUV instruments have been flown in sun-synchronous orbits, which provide measurements
up to ±81° latitude. However, each satellite has drifted from its original Equator-crossing time
(typically 1340-1400 LT), so that the local time of measurements at any specific latitude varies
over the lifetime of the instrument.

The consistent design of all SBUV/2 instruments allows the same PMC detection algorithm to be
used with each data set, and the overlapping lifetime of these instruments (Figure 1) enables the
creation of a merged data set long enough to be used for trend studies. Development and updates
to this data set have been published by DeLand et al. (2006), DeLand et al. (2007), Shettle et al.
(2009), and DeLand and Thomas (2015).  Additional recent studies of long-term PMC behavior
that use the SBUV PMC data set include Hervig and Stevens (2014), Berger and Lübken (2015),
Hervig et al. (2016), Fiedler et al. (2017), Kuilman et al. (2017), and von Savigny et al. (2017).

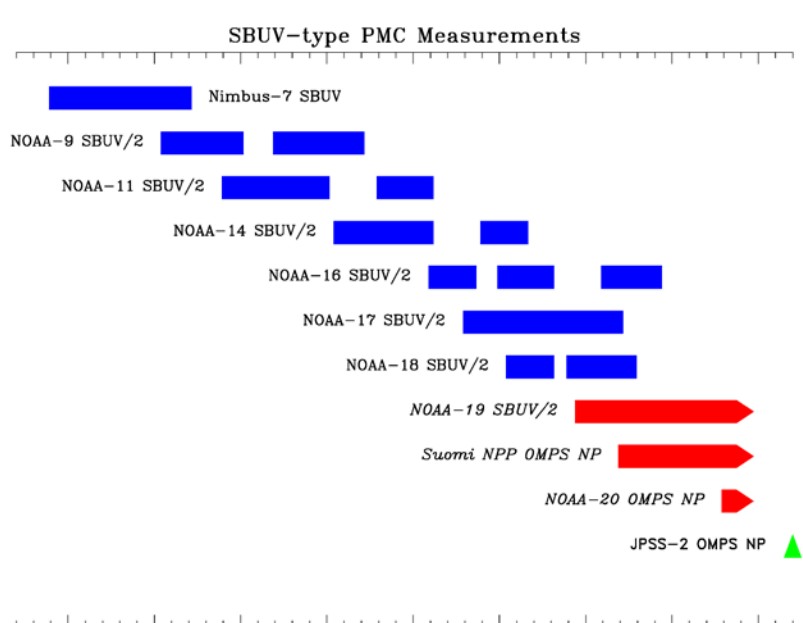



**Figure 1.**  Timeline of SBUV instrument measurements used for PMC analysis.

Blue color indicates inactive instruments.  Arrowheads and red color indicate

active instruments.  Green color indicates planned instrument.  Gaps for many

SBUV/2 instruments reflect satellite drift into a near-terminator orbit where the

current PMC detection algorithm does not function well.


The last SBUV/2 instrument is now flying on the NOAA-19 spacecraft.  Its sun-synchronous
orbit has drifted significantly from its original 1340 LT ascending node Equator-crossing time
(current Equator-crossing time = 1615 LT), which will interrupt the ability to extract PMC
information in 2019 or 2020 due to the decrease in solar zenith angle range available for daytime
measurements.  Fortunately, the SBUV measurement concept is being continued by the Ozone
Mapping and Profiling Suite (OMPS) Nadir Profiler (NP) instrument (Seftor et al., 2014), which
is now orbiting on two satellites.  This paper will describe updated PMC trends that extend the
work of DeLand and Thomas (2015), including the addition of OMPS NP data to the 40-year
merged SBUV PMC dataset.  Section 2 of this paper presents PMC occurrence frequency and ice
water content (IWC) results from concurrent measurements by the NOAA-19 SBUV/2 and
Suomi National Polar-orbiting Partnership (S-NPP) OMPS NP instruments.  We then use these
data in Section 3 to extend the long-term IWC trend analysis of DeLand and Thomas (2015) into
2018, thus creating a 40-year merged PMC data set.  We find that separating this data set into
two sections, with a break point selected in 1998 (as described in that section), provides an
effective characterization of PMC behavior throughout this long data record.

**2. OMPS NP Data**

The OMPS NP instrument was developed to provide ozone data that are consistent with the
SBUV/2 series of instruments (Flynn et al., 2014).  The first OMPS NP instrument was launched
on the Suomi National Polar-orbiting Partnership (S-NPP) satellite on 28 October 2011, and
began collecting regular data in January 2012.  It makes hyperspectral measurements covering
the 250-310 nm spectral region, with a sampling of approximately 0.6 nm.  We utilize radiance
measurements interpolated to the five shortest SBUV/2 wavelengths (nominally 252.0, 273.5,
283.1, 287.6, 292.3 nm) to provide continuity with the current SBUV PMC detection algorithm.
Potential retrieval improvements based on a different wavelength selection will be explored in
the future.  The NP instrument uses a larger field of view (250 km x 250 km at the surface)
compared to a SBUV/2 instrument.  We will show that this difference does not affect the ability
of the NP instrument to track seasonal PMC behavior.

The only revision implemented to the SBUV PMC detection algorithm for OMPS NP is to derive
a solar zenith angle-dependent detection threshold in albedo that is based on NP end-of-season
measurements, rather than SBUV measurements.  This update ensures that any change in
background variability introduced by the larger NP field of view is addressed.  Figure 2 shows
the NP threshold function derived as a quadratic fit to data taken during 11-31 August 2012,
when very few PMCs are typically detected in SBUV-type data.  Note that for a nadir-viewing
instrument such as NP, the solar zenith angle (SZA) is equivalent to the supplement of the
scattering angle (SCA), i.e. $SZA = 180° - SCA$.  The SBUV/2 threshold function determined by
DeLand and Thomas (2015) is shown for comparison, where an empirical scaling factor of 1.6 is
also applied to eliminate "false positive" PMC detections at the start and end of the PMC season.
These functions differ slightly at low solar zenith angle, but are almost identical at SZA > 50°.
The uncertainty in this detection threshold is approximately $\pm 3 \times 10^{-6}$ sr$^{-1}$. This value is driven by
albedo fluctuations due to meridional variations in stratospheric ozone, since the magnitude of
the backscattered albedo at wavelengths used for PMC detection (250-290 nm) is dominated by
ozone absorption.

DeLand and Thomas (2015) noted that fluctuations in 252 nm albedo (caused by lower signal-to-
noise performance relative to other wavelengths) could lead to unrealistically faint scenes being
identified as PMC detections. They implemented an additional requirement for trend analysis
that the albedo residual at 273 nm be greater than $3 \times 10^{-6}$ sr$^{-1}$ at all SZA. Converting this albedo
value into IWC gives an effective threshold that ranges between 35-40 g km$^{-2}$, as shown in
Figure 2. This value is consistent with the IWC threshold of 40 g km$^{-2}$ determined by Hervig
and Stevens (2014) for their analysis of SBUV PMC data. It is important to note that additional
tests focusing on spectral dependence of the albedo residuals are also applied to positively
identify any sample as a PMC.

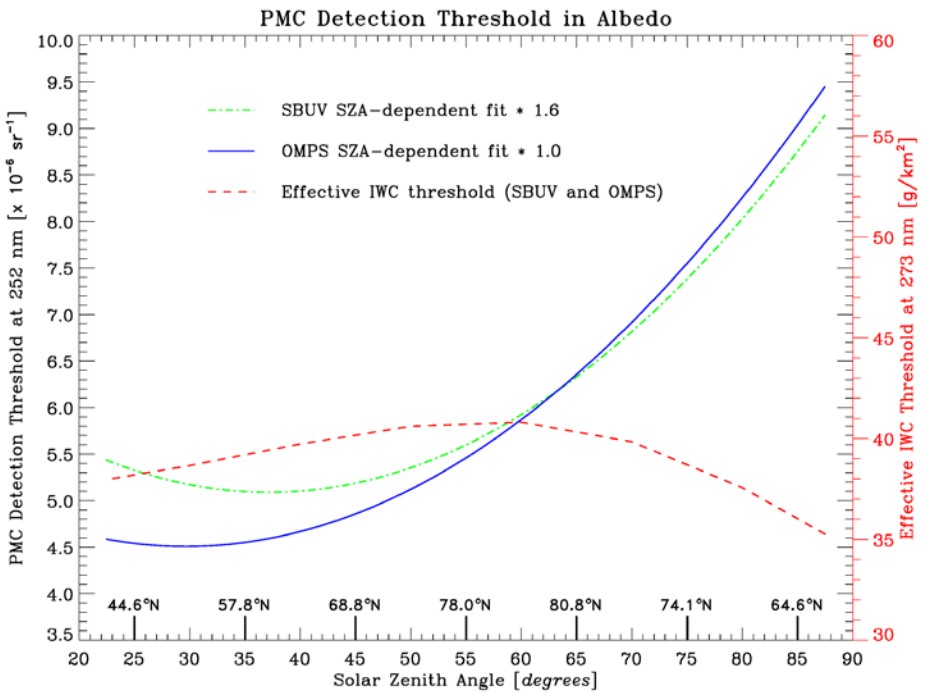



**Figure 2.** PMC detection threshold functions plotted *vs.* solar zenith angle

(SZA). The quadratic fit in SZA used by DeLand and Thomas (2015) for

SBUV/2 processing, derived from NOAA-18 data taken in 2007 days 222-242, is

shown as the dot-dash line (green). The quadratic fit in SZA used for OMPS NP

data in this paper, derived from S-NPP data taken in 2012 days 222-242, is shown

as the solid line (red). The local time sampling is very similar (1335 LT Equator-

crossing time for NOAA-18, 1340 LT Equator-crossing time for S-NPP). The

effective IWC threshold (described in the text) is shown as the dashed line (red),

and referenced to the scale on the right-hand Y-axis. Nominal latitude values for

June 21 are identified on the bottom of the plot.


Figure 3 illustrates the PMC detection results obtained for a single day of S-NPP OMPS NP data.
The top panel shows the individual albedo values at 273.7 nm for all 14 orbits. These values are
tightly grouped in SZA because OMPS NP uses a measurement sequence that begins at the
Southern Hemisphere terminator (SZA = 90°) for each orbit, and continues in 38 second
increments throughout the day side of the orbit. There is very little change in latitude for the
terminator crossing during a single day, which leads to repeatable sample latitudes on the same
time scale, although the terminator crossing location does shift over the course of the PMC
season. Samples identified as PMCs are shown as squares. The bottom panel shows the albedo
residual (difference between observation and background fit) for the same date. Note that an
arbitrary PMC would be expected to have a stronger signal in albedo at lower scattering angles
(= higher SZA) due to the forward scattering peak of the small ice particles (DeLand et al., 2011;
Lumpe et al., 2013). We do not adjust the observed albedo values with any assumed phase
function before applying our PMC detection algorithm, so the SZA dependence of the albedo
threshold shown in Figure 2 represents a method to incorporate this sensitivity in our analysis.
The spread of the non-PMC albedo residual values due to both longitudinal and along-track
ozone variability is $\sim$3-5x10$^{-6}$ sr$^{-1}$ at latitudes less than approximately 60° (SZA < 40°), and
increases slightly at higher latitudes where ozone variability is greater.

Some improvement in the detection of faint PMCs using this algorithm is possible when
measurements are spaced closely enough in time that the background fit can be calculated
separately for each orbit, thus eliminating the effects of longitudinal variations in ozone.
DeLand et al. (2010) used this approach with Aura OMI data, which have a 13 km along-track
sampling.  Even with these data, though, non-PMC samples at low latitude still fluctuate by
$\pm 3 \times 10^{-6}$ sr$^{-1}$ around the background fit (see their Figure 5).  The minimum PMC detection
threshold for nadir-only measurements is thus higher than the level available to an instrument
such as CIPS that incorporates multiple viewing angles, and the accompanying phase function
information, to separate clouds from background samples.

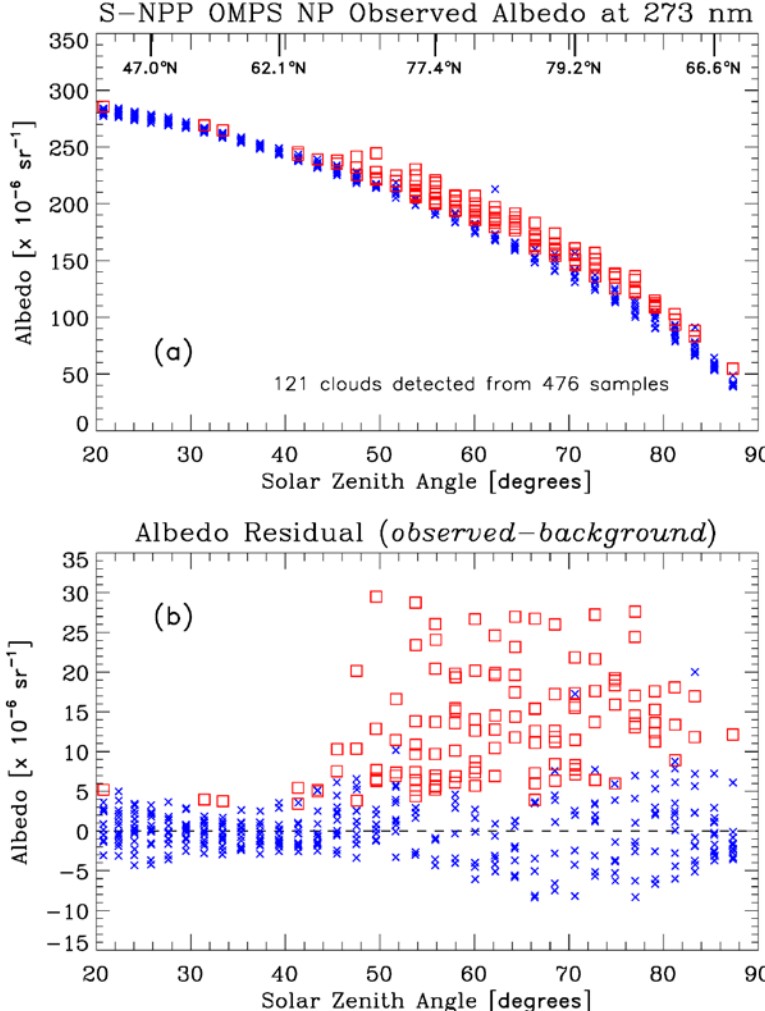



**Figure 3.** (a) S-NPP OMPS NP 273 nm albedo values for all measurements on
2018 day 189. Squares (red) indicate measurements identified as PMCs by the
detection algorithm. Crosses (blue) indicate non-PMC samples. Tick marks (top
X-axis) show approximate latitudes corresponding to selected solar zenith angle
values. (b) 273 nm albedo residuals (observed-background fit) for the
measurements shown in panel (a). PMC detections are indicated by squares.

The fluctuations in the background data represent a significant component of the uncertainty in
PMC albedo for any individual detection. Using the uncertainty analysis described in DeLand et
al. (2003), we estimate that this term gives a value of $\sim 2 \times 10^{-6}$ $sr^{-1}$ for PMC albedo uncertainty at
273 nm for most latitudes. There is also a possible bias in PMC albedo due to the presence of
faint (but otherwise valid) clouds in the background fit calculation. Examination of the seasonal
variation in background fit at fixed SZA values suggests that there is no bias at latitudes less than
$\sim 70°$, increasing to a possible bias of $\sim 2\text{-}3 \times 10^{-6}$ $sr^{-1}$ at 75°-81° latitude.

We next compare S-NPP OMPS NP PMC occurrence frequency and ice water content (IWC)
seasonal average results to concurrent NOAA-19 SBUV/2 PMC results for seven Northern
Hemisphere (NH) and six Southern Hemisphere (SH) PMC seasons from NH 2012 through NH
2018. IWC values are derived from PMC albedo values using the albedo-ice regression (AIR)
approach described in DeLand and Thomas (2015). This approach parameterizes output from a
coupled general circulation model and microphysical model to create linear fits for IWC as a
function of PMC albedo at multiple scattering angles. Thomas et al. (2018) present a more
extensive description of the AIR approach. Figures 4-6 show these comparisons for the latitude
bands 50°-64°, 64°-74°, and 74°-82° respectively. We define the length of each season as [-20
days since solstice (DSS), +55 DSS] for PMC trend analysis, following the discussion presented
in DeLand and Thomas (2015). All averages use both ascending node and descending node data
where available. Since most of the uncertainty in IWC values comes from random variations in
albedo, as discussed in DeLand et al. (2007), we show the standard error [(standard deviation) /
(number of clouds)$^{1/2}$] of each seasonal average IWC value in the right-hand panels. The
nominal SZA and local time values for these averages are given in Table 1, as well as the total
number of samples and PMCs detected. The two instruments agree very well in both absolute
level and interannual variability for both quantities in each latitude band.  The occurrence
frequency difference between instruments in the NH 2016 season at 64°-74° N (Figure 5(a)) is
anomalous, and does not appear in IWC results for the same season (Figure 5(b)).  We believe
that the S-NPP OMPS result is the outlier in this case.  We are satisfied that S-NPP OMPS NP
data can be added to the SBUV PMC data set to continue the long-term record in a consistent
manner.

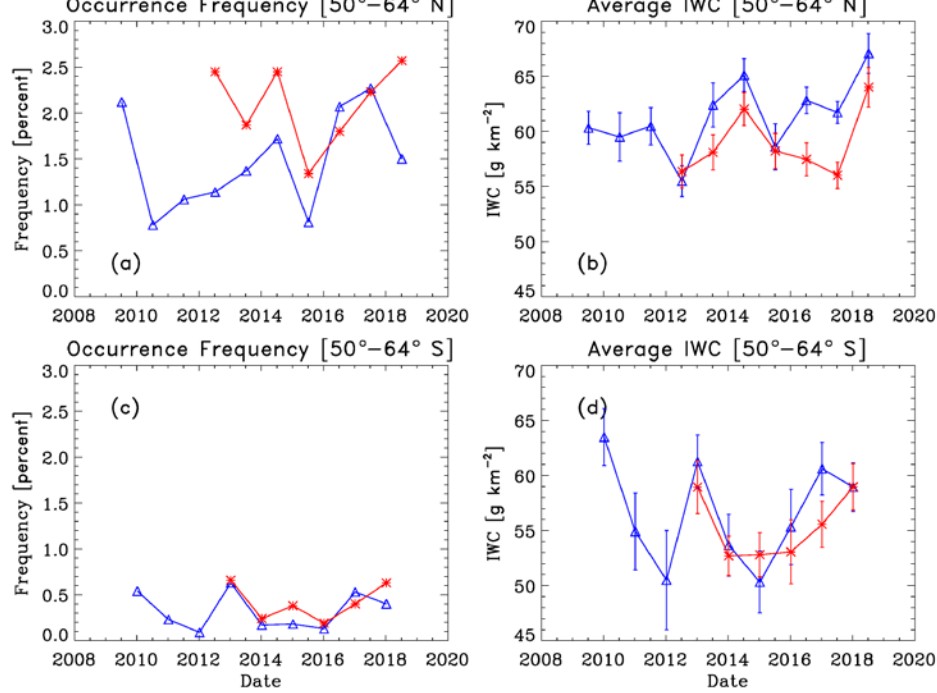



**Figure 4.**  Season average PMC occurrence frequency and ice water content data

at 50°-64° latitude.  Blue = NOAA-19 SBUV/2, red = S-NPP OMPS.  Left side =

occurrence frequency [percent], right side = IWC [g km$^{-2}$].  Top row = Northern

Hemisphere, bottom row = Southern Hemisphere.  Average SZA and local time

values for each instrument during each season are listed in Table 1.


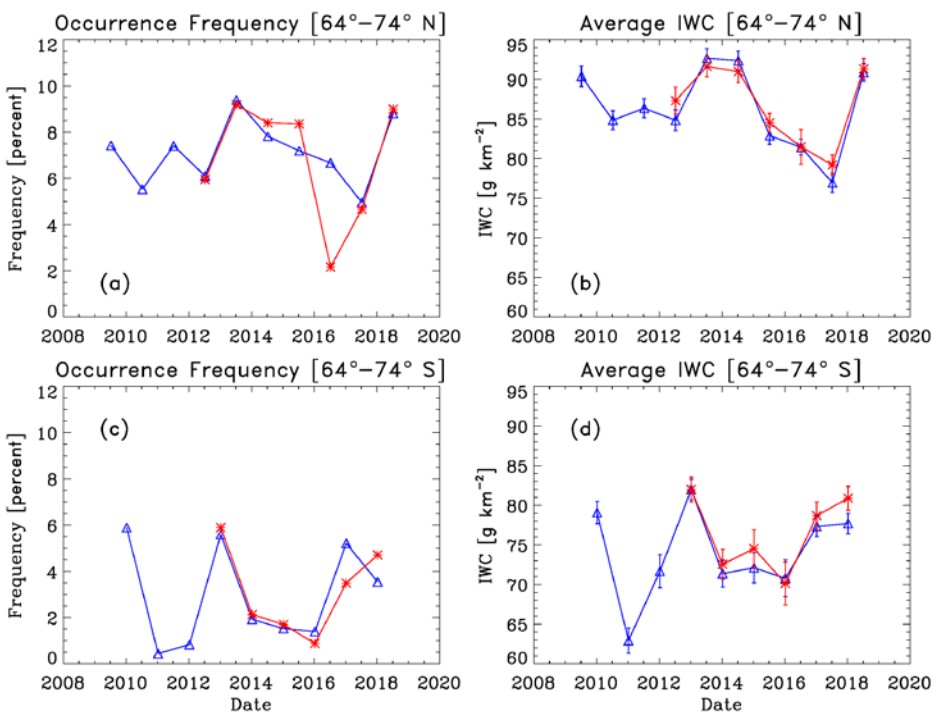



**Figure 5.** Season average occurrence frequency and IWC data at 64°-74°
latitude. Identifications are as in Figure 4.

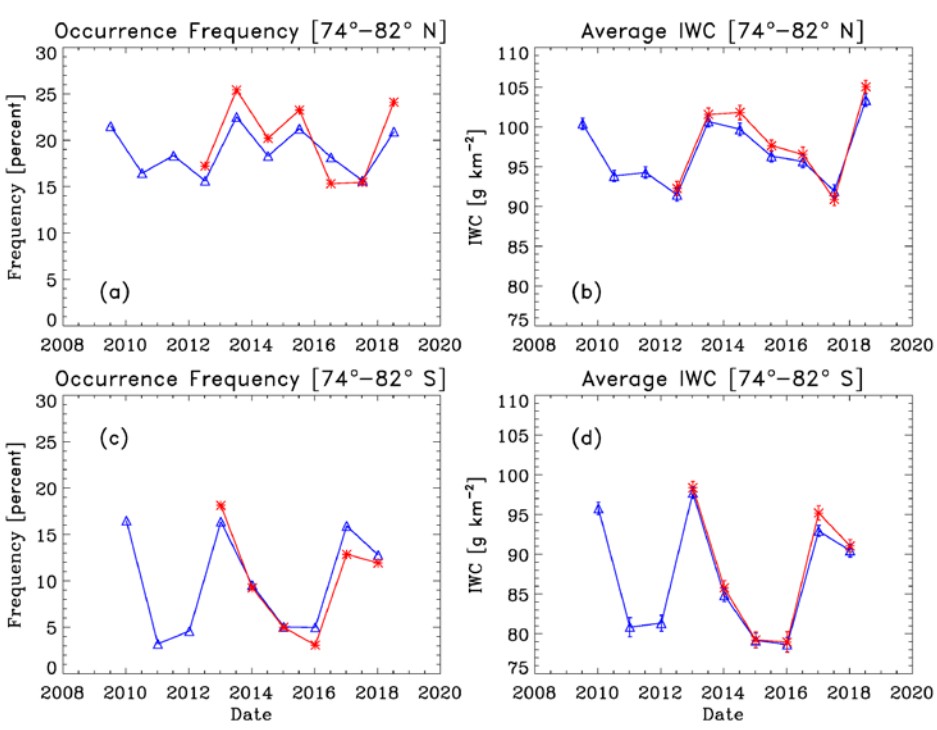



**Figure 6.** Season average occurrence frequency and IWC data at 74°-82°

latitude. Identifications are as in Figure 4.


Some model results (e.g. Stevens et al. (2017) show significantly higher PMC occurrence
frequency values (factor of 5-10) for clouds that exceed the nominal SBUV IWC threshold of 40
g km$^{-2}$. However, there are important factors that should also be considered in such
comparisons. The use of an idealized PMC formation mechanism based on bulk thermodynamic
properties (Hervig et al., 2009) in the model calculations will yield a high amount of PMCs in
many situations. Using only results from the peak of the diurnal cycle at 4 LT, as chosen by
Stevens et al. (2017), will produce substantially higher frequency values than those determined
in this paper by averaging both ascending node (10-13 LT) and descending node (3-5 LT) data.
Stevens et al. (2017) calculate seasonal averages using only the core of the NH season in July
(DSS = [+10,+40]), which can give a factor of two or more higher occurrence frequency values
compared to the longer season definition used in this paper. Fiedler et al. (2017) show seasonal
average occurrence frequency values between 3-12% during 1997-2015 for strong clouds (most
similar to SBUV detections) observed by the Arctic Lidar Observatory for Middle Atmosphere
Research (ALOMAR) lidar at 69°N, which is similar to the SBUV/2 and OMPS frequency
values shown in Figure 5(b) for 64°-74°N. Schmidt et al. (2018) show seasonal occurrence
frequency values from the Mesospheric Ice Microphysics And tranSport (MIMAS) model at
69°N that are consistent with ALOMAR results for strong clouds.

Hervig and Stevens (2014) suggest that there may be a bias in the SBUV background calculation,
based on their analysis of the number of selected (as PMC) and non-selected SBUV samples
above a constant albedo threshold (5x10$^{-6}$ sr$^{-1}$ at 252 nm). This approach is not correct at low
scattering angle (high SZA), since our actual threshold for the V3 product that they consider is
determined by the SZA-dependent function shown in Figure 2. We have examined the seasonal
variation of our background fit at fixed SZA values. We find no evidence for background error
at 65°-70° latitude, but a possible high bias during the core of the PMC season of ~2-3x10$^{-6}$ sr$^{-1}$
at 75°-81° latitude. It is difficult to determine whether this result represents faint PMCs that are
"embedded" in the background data and not currently identified (thus representing a bias), or
whether it indicates increased stratospheric ozone variability for this latitude and time of year.

The nadir viewing geometry of SBUV and OMPS means that only bright PMCs, composed of
relatively large ice particles, will be detected above the Rayleigh scattering background.  Our
SBUV PMC detection algorithm does not yield particle size, but estimates can be made based on
other methods.  Bailey et al. (2015) state that CIPS detects almost 100% of PMCs with a mean
particle radius greater than 30 nm, based on a nominal brightness of $2x10^{-6}$ sr$^{-1}$ and a 90°
scattering angle.  Lumpe et al. (2013) quote a CIPS detection threshold of IWC > 10 g km$^{-2}$.  The
minimum SBUV IWC value is ~40 g km$^{-2}$ based on our albedo threshold (Figure 2), which is
consistent with the empirical result derived by Hervig and Stevens (2014).  They find a median
particle size of $r_m \approx 30$ nm for their long-term analysis of the SBUV record, using only data
measured between 9-15 LT, compared to a median size of $r_m \approx 38$ nm for SBUV measurements
used in SOFIE-SBUV coincidence studies.  In addition, SBUV PMCs are only observed at
scattering angles greater than 90°, which will give a lower PMC brightness for a given particle
size compared to the CIPS definition.  These factors suggest that SBUV and OMPS instruments
only detect PMCs with mean particle radius > 35-40 nm.  Stevens et al. (2017) calculated daily
average IWC during July 2009 as a function of latitude, using output from the NOGAPS-
ALPHA forecast-assimilation system and the Hervig et al. (2009) 0-D model to create IWC
values from these data.  When they apply a threshold of IWC > 40 g km$^{-2}$, their zonal average
results are approximately 20-30% greater than the NOAA-19 SBUV/2 seasonal average values
for NH 2009 shown in Figure 4(b), Figure 5(b), and Figure 6(b).  Possible causes for this
difference include the use of July-only averages compared to the longer season defined in this
paper, the averaging of model results at all local times compared to the specific local time of the
measurements (plus local time adjustment described in Section 3), and the different methods
used to create IWC values.

**3. Trend Update**

Our analysis of long-term trends in SBUV PMC data follows the approach presented in DeLand
et al. (2007), and updated by DeLand and Thomas (2015).  We use IWC as our key variable for
trend analysis because it provides a way of minimizing the effects due to variations in scattering
angle caused by the drifting orbit of many SBUV instruments.  The seasonal average IWC values
do not incorporate frequency variation, i.e. only samples with a positive PMC detection are used.
This choice reduces the magnitude of interannual fluctuations, particularly in the SH where
SBUV occurrence frequency results are more variable, and allows us to focus on a quantity
[IWC derived from measured albedo] that we feel most confident in evaluating.  Long-term
trends in SBUV PMC frequency were derived by Shettle et al. (2009), and are also considered in
Pertsev et al. (2014).  As in our earlier publications, we use a multiple regression fit of the form

$$X_{fit}(latitude,t) = A(latitude)*F_{Ly\alpha}(t) + B(latitude)*(t-1979) + C(latitude) \qquad [1]$$

where $F_{Ly\alpha}(t)$ is the composite solar Lyman alpha flux dataset available from the LASP
Interactive Solar Irradiance Data Center (LISIRD) and averaged over the appropriate NH or SH
season.  We assess the quantitative significance of the trend term by calculating a 95%
confidence limit as described in DeLand et al. (2007), using a method presented by Weatherhead
et al. (1998) that accounts for periodicity auto-correlation in addition to the fit uncertainty.

The orbit drift experienced by most SBUV instruments causes significant changes in local time
sampling for any selected latitude band over our 40-year PMC data record.  Since lidar
measurements show significant local time dependence in PMC properties (e.g. Chu et al., 2006;
Fiedler et al., 2011), it must be addressed for trend analysis.  One approach is to define a limited
local time range that is always sampled (Hervig and Stevens, 2014;  Hervig et al., 2016).
However, this reduces the amount of data available (only ascending or descending node data can
be used except near 81° latitude), and the time range must be adjusted for different latitude
bands.  We have chosen to apply a diurnal harmonic function to normalize all observations to a
single local time (11 hr LT).  The derivation of this function from SBUV data is described in
detail by DeLand and Thomas (2015).

$$F(t) = A_0 + A_{24}*cos[(2\pi/24)*(t-\varphi_{24})] \qquad [1]$$

$\qquad A_0 = 110 \qquad A_{24} = 8 \qquad \varphi_{24} = 2\ hr \qquad F_{norm}(t) = F(t)/F(11\ h)$

The SBUV local time dependence created by DeLand and Thomas (2015) and used in this paper
was based on observations at a limited set of local times.  A single diurnal function with a
maximum/minimum ratio of ~1.15 was derived for use at all latitudes.  This function was shown
to have a similar shape, but somewhat smaller amplitude, than lidar-based functions determined
by Fiedler et al. (2011) and Chu et al. (2006).  Recent model results provide local time
dependence functions at different latitude bands for multiple levels of IWC threshold.  Stevens et
al. (2017) determined a maximum/minimum ratio of ~1.4 for the IWC variation (no frequency
weighting) at 90°N in July 2009, using only model PMCs with $IWC > 40$ g km$^{-2}$.  This ratio
decreases slightly at lower latitudes (55°N, 60°N) and higher latitude (80°N).  Schmidt et al.
(2018) created IWC local time variations from 35 years of model output (1979-2013) for the
three broad latitude bands used in this paper (50°-64°N, 64°-74°N, 74°-82°N) and three
threshold levels ($IWC > 0, > 10, > 40$ g km$^{-2}$).  The "strong" cloud results ($IWC > 40$) all show
greater maximum/minimum ratios than the SBUV function, with values increasing from 1.3 at
50°-64°N to 2.1 at 74°-82°N.  This latitude dependence differs from Stevens et al. (2017) and the
Aura OMI results shown by DeLand et al. (2011), where the local time amplitude decreases at
higher latitude.  We have not yet investigated the impact of using one of these model-based local
time dependence functions in our trend analysis.

We define the duration of the PMC season for our trend analysis as DSS = [-20,+55] to fully
capture interannual variations (DeLand and Thomas, 2015).  We have also examined the impact
of limiting our season to a "core" range of DSS = [+10,+40] to correspond to July in NH summer
and January in SH summer, as used in other studies.  The numerical values calculated for the
trend term do change slightly for each latitude band, as expected.  However, the determination of
whether a trend result exceeds the 95% confidence level defined above does not change for any
latitude band with the use of core seasons.  This implies that our conclusions regarding long-term
behavior are robust.

We created a merged SBUV PMC IWC data set for each season and latitude band, using an
adaptation of the "backbone" method of Christy and Norris (2004) as discussed by DeLand et al.
(2007).  An advantage of this method is that it easily accommodates the addition of new
instruments such as S-NPP OMPS NP to the overall PMC data set. Normalization adjustment
values for each SBUV and OMPS instrument derived from a fit at 50°-82° latitude are applied
consistently at all latitude bands. The adjustment values for merging derived in this work are
slightly different than those derived by DeLand and Thomas (2015) because the composition of
the overall data set has changed, even though the original V4 PMC data sets for each instrument
as described in that paper have not changed. Almost all adjustment values are still less than 3%
of the seasonal average IWC (e.g. 0.97-1.03), and most of the changes in the adjustment values
determined for this paper relative to DeLand and Thomas (2015) are smaller than ±0.01.
Performing the trend analysis with no merging adjustments does not change the results for
exceeding the 95% confidence level in any latitude band, similar to the core season analysis
described above. We have not evaluated this data set for the possibility of longitudinally
dependent trends, as was done by Fiedler et al. (2017).

Berger and Lübken (2011) calculated long-term trends in PMC scattered brightness by coupling
3-D atmospheric model runs (driven by lower atmosphere reanalysis data) with a microphysics
module that simulates PMC ice particle formation. They found that the long-term trend in
mesospheric temperature at 83 km changed from negative to positive in the late 1990s, and
suggested that this change was forced by an increase in stratospheric ozone and its subsequent
impact on middle atmospheric heating rates. This implies that a single linear segment is not the
best way to represent trends since 1978. Since PMC properties are expected to be very
responsive to mesospheric temperature changes, DeLand and Thomas (2015) followed this
guidance and calculated their PMC trends in two segments, with a break point in 1998. We
follow the same approach here and calculate multiple regression fits for two time segments,
covering 1979-1997 and 1998-2018 respectively.

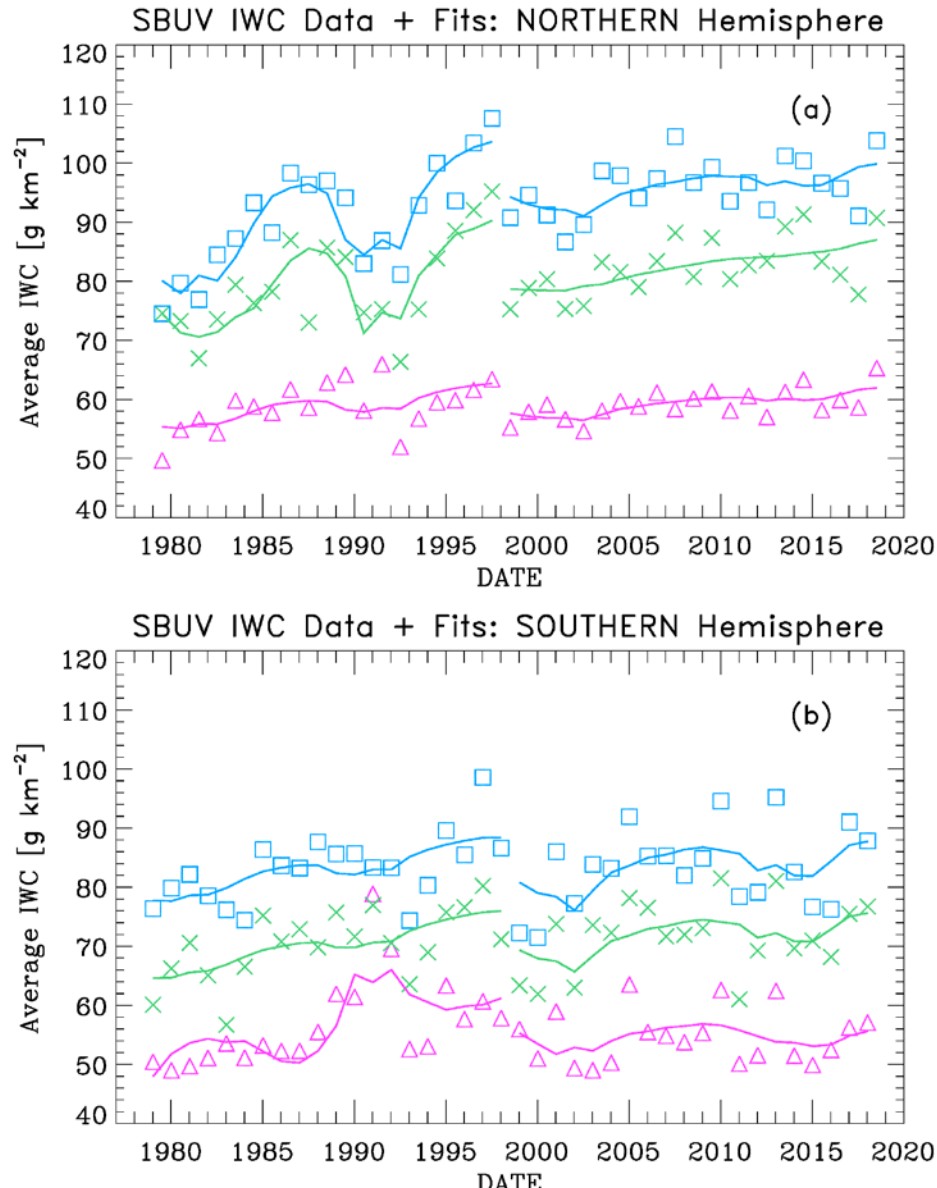

**Figure 7.** (a) SBUV merged seasonal average IWC values for three different latitude bands: 50°-64° N (purple triangles), 64°-74° N (green crosses), 74°-82° N (blue squares). The solid lines show multiple regression fits to the data for the periods 1979-1997 and 1998-2018. (b) SBUV merged seasonal average IWC values for 50°-64° S, 64°-74° S, and 74°-82° S. The solid lines show fits for the periods 1979-1997 and 1998-2018.

The results of these fits are shown in Figure 7, and presented numerically in Tables 2 and 3.
Note that a negative sign for the solar activity term implies an anti-correlation, i.e. an increase in
solar activity corresponds to a decrease in IWC. This behavior has been explained by variations
in solar ultraviolet irradiance, which causes higher temperatures and lower water vapor
abundance during solar maximum periods (Garcia, 1989). The trend term and solar term results
for each hemisphere are discussed below.

a. NH trend term. These results are significant at the 95% confidence level (as defined in

the previous paragraph) for all latitude bands in both segments, although the trend values for
segment 2 (1998-2018) are smaller than those derived by DeLand and Thomas for a shorter
period (1998-2013). The changes in this term do not exceed the ±1 σ uncertainty of the current
fit results in any latitude band, as shown in Table 2(b).

b. SH trend term. These values exceed our 95% confidence limit in segment 1, consistent

with DeLand and Thomas (2015). However, the segment 2 trend values are a factor of 2-4
smaller than those derived by DeLand and Thomas (2015), and no latitude band reaches the 95%
confidence limit. We discuss this result further in part (d). Note that the difference between
hemispheres has been explained by Siskind et al. (2005) to be caused by higher SH mesospheric
temperatures, making SH PMCs more sensitive to small temperature changes.

c. NH solar term. These values are significant at the 95% level for most latitude bands

for segment 1, consistent with DeLand and Thomas (2015). Phase lag values of 0.5-1.0 years are
found, consistent with previous analysis of SBUV PMC data. The fit values for segment 2 are
smaller than those derived for segment 1 by as much as a factor of seven, depending on latitude
band, and in general are not larger than the ±1 σ uncertainty. This lack of response to solar
activity in recent years has also been identified in ALOMAR lidar PMC data (Fiedler et al.,
2017) and AIM CIPS data (Siskind et al., 2013).

410       d. SH solar term. These values poleward of 64° latitude are smaller than the ±1 σ

uncertainty in segment 1, but become 2-3 times larger and exceed the 95% significance level in
segment 2. However, note also that the correlation coefficient for this term is quite low ($r =$
0.19). We speculate that during segment 2, the multiple regression fit algorithm is assigning
some of the greater interannual variability in SH data to the solar activity term. The large
positive solar term at 50°-64° S is driven by higher IWC values in the 1990-1991 and 1991-1992
seasons. In this latitude band, only 10-20 clouds are detected from 6000-8000 samples during
the entire season in some years, as shown in Table 1.  Fluctuations in only a few samples can
thus have a significant impact in such seasons.

These result illustrate the need for caution in interpreting the results of using a periodic term
based on solar variability in a regression fit that covers less than two full solar cycles for a single
segment, since variations in a small number of data points near the end of the period can have a
substantial impact.  However, the large IWC values observed in the recent NH 2018 PMC season
did not significantly change the NH solar activity term for this segment.  Both the source of the
hemispheric difference in solar activity response and the source of the derived phase lag in the
NH are not understood.

**4. Conclusion**

We have shown that OMPS NP measurements can be used successfully to continue the long
PMC data record created from SBUV and SBUV/2 instruments.  When we use S-NPP data to
extend our merged PMC data set through the NH 2018 season, we find smaller trends in IWC in
both hemispheres since 1998 compared to the results shown by DeLand and Thomas (2015).
The NH trends continue to be significant at the 95% confidence level, while the SH trends are
now slightly smaller than this threshold.  The calculated sensitivity to solar activity during 1998-
2018 is a factor of three to six smaller than the 1979-1997 result for NH data poleward of 64° N.
However, the solar activity sensitivity for SH data increases by a factor of three to four for the
1998-2018 period, and becomes statistically significant at all latitudes.  We will continue to
investigate possible causes for this change in behavior and hemispheric discrepancy.

A second OMPS NP instrument was launched on the NOAA-20 (formerly JPSS-1) satellite in
November 2017, and is now collecting regular data.  Three more OMPS NP instruments are
scheduled for launch on JPSS satellites at regular intervals through approximately 2030.  All of
the satellites carrying OMPS NP instruments will be kept in an afternoon equator-crossing time
sun-synchronous orbit, so that orbit drift (which has impacted all SBUV/2 instruments) will not
affect the ability to retrieve PMC information.  We therefore anticipate extending the continuous
SBUV PMC data record to 60 years to support long-term climate studies.

**Data Availability.** Daily IWC data for all SBUV instruments during every season are available on-line at https://sbuv2.gsfc.nasa.gov/pmc/v4/. A text file describing the contents of these files is also provided. Solar Lyman alpha flux data is available at http://lasp.colorado.edu/lisird/.


**Author Contributions.** MD processed the SBUV and OMPS PMC data, conducted the regression fit analysis, and wrote the primary manuscript. GT reviewed and edited the manuscript.


**Acknowledgements.** We greatly appreciate the continuing efforts of Larry Flynn and many other people at NOAA STAR to provide high quality SBUV/2 and OMPS NP data that enable the creation of our PMC product. We thank the reviewers for their comments that have improved the content of this paper. M. T. DeLand was supported by NASA grant NNH12CF94C. G. Thomas was supported by the NASA AIM mission, which is funded by NASA's Small Explorers Program under contract NAS5-03132.

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

**Table 1(a)**

Statistics for NOAA-19 SBUV/2 Northern Hemisphere PMC Seasons, 2009-2018

| Latitude | Season | Ntotal | Ncloud | LTasc | LTdesc | SCAasc | SCAdesc |
|---|---|---|---|---|---|---|---|
| 50°-64° N | 2009 | 8964 | 190 | 12.9 | 3.0 | 142.7° | 93.5° |
| | 2010 | 8624 | 67 | 12.7 | 2.9 | 143.4° | 93.1° |
| | 2011 | 8525 | 90 | 12.6 | 2.8 | 143.7° | 92.9° |
| | 2012 | 8366 | 95 | 12.6 | 2.8 | 143.7° | 92.9° |
| | 2013 | 8661 | 119 | 12.7 | 2.8 | 143.4° | 93.1° |
| | 2014 | 8912 | 153 | 12.9 | 3.1 | 142.8° | 93.5° |
| | 2015 | 9683 | 78 | 13.3 | 3.4 | 141.6° | 94.2° |
| | 2016 | 11019 | 228 | 13.7 | 3.8 | 139.4° | 95.2° |
| | 2017 | 13639 | 309 | 14.4 | 4.4 | 135.7° | 96.6° |
| | 2018 | 16364 | 246 | 15.1 | 5.1 | 130.5° | 100.1° |
| 64°-74° N | 2009 | 11764 | 873 | 12.3 | 3.5 | 132.0° | 98.5° |
| | 2010 | 11654 | 645 | 12.0 | 3.3 | 132.2° | 98.0° |
| | 2011 | 11582 | 858 | 11.9 | 3.2 | 132.2° | 97.7° |
| | 2012 | 11380 | 694 | 11.9 | 3.2 | 132.2° | 97.7° |
| | 2013 | 11647 | 1094 | 12.0 | 3.3 | 132.1° | 98.0° |
| | 2014 | 11850 | 927 | 12.2 | 3.6 | 132.1° | 98.6° |
| | 2015 | 12273 | 882 | 12.6 | 3.9 | 131.8° | 99.8° |
| | 2016 | 12543 | 836 | 13.0 | 4.4 | 131.2° | 101.4° |
| | 2017 | 12567 | 662 | 13.6 | 5.0 | 129.6° | 104.2° |
| | 2018 | 12758 | 1124 | 14.4 | 5.8 | 127.2° | 108.2° |
| 74°-82° N | 2009 | 15264 | 3286 | 9.9 | 5.3 | 120.5° | 108.2° |
| | 2010 | 15349 | 2525 | 9.7 | 5.1 | 120.2° | 107.6° |
| | 2011 | 15276 | 2803 | 9.6 | 5.0 | 120.1° | 107.4° |
| | 2012 | 15008 | 2345 | 9.6 | 5.0 | 120.1° | 107.4° |
| | 2013 | 15223 | 3428 | 9.7 | 5.1 | 120.1° | 107.6° |
| | 2014 | 15134 | 2769 | 9.9 | 5.4 | 120.5° | 108.3° |
| | 2015 | 15144 | 3216 | 10.3 | 5.8 | 121.2° | 109.6° |
| | 2016 | 15084 | 2740 | 10.8 | 6.3 | 121.7° | 111.1° |
| | 2017 | 14944 | 2339 | 11.4 | 7.0 | 121.9° | 112.8° |
| | 2018 | 15066 | 3150 | 12.2 | 7.7 | 121.9° | 115.1° |

Ntotal = Number of samples in latitude band during season (DSS = [-20,+55])

Ncloud = Number of PMC detections

LTasc = Average local time for ascending node samples [hr]

LTdesc = Average local time for descending node samples [hr]

SCAasc = Average scattering angle for ascending node samples

SCAdesc = Average scattering angle for ascending node samples

**Table 1(b)**

Statistics for NOAA-19 SBUV/2 Southern Hemisphere PMC Seasons, 2009-2018

| Latitude | Season | Ntotal | Ncloud | LTasc | LTdesc | SCAasc | SCAdesc |
|----------|--------|--------|--------|-------|--------|--------|---------|
| 50°-64° S | 2009-2010 | 8355 | 45 | 14.7 | – | 133.3° | – |
| | 2010-2011 | 8321 | 19 | 14.5 | – | 134.4° | – |
| | 2011-2012 | 8134 | 7 | 14.5 | – | 134.7° | – |
| | 2012-2013 | 8270 | 52 | 14.5 | – | 143.7° | – |
| | 2013-2014 | 8259 | 14 | 14.7 | – | 143.4° | – |
| | 2014-2015 | 8363 | 15 | 15.0 | – | 142.8° | – |
| | 2015-2016 | 8353 | 11 | 15.4 | – | 141.6° | – |
| | 2016-2017 | 8268 | 44 | 16.0 | – | 139.4° | – |
| | 2017-2018 | 8336 | 33 | 16.7 | – | 135.7° | – |
| 64°-74° S | 2009-2010 | 8479 | 499 | 15.4 | 16.6 | 122.8° | 93.9° |
| | 2010-2011 | 8468 | 37 | 15.2 | 22.9 | 123.5° | 94.0° |
| | 2011-2012 | 8302 | 69 | 15.2 | 23.6 | 123.7° | 94.0° |
| | 2012-2013 | 8433 | 471 | 15.2 | 23.5 | 123.5° | 94.0° |
| | 2013-2014 | 8383 | 161 | 15.4 | 18.8 | 122.6° | 93.9° |
| | 2014-2015 | 8542 | 130 | 15.7 | 7.9 | 121.3° | 93.9° |
| | 2015-2016 | 8709 | 121 | 16.1 | 0.5 | 119.3° | 93.9° |
| | 2016-2017 | 9051 | 472 | 16.7 | 1.1 | 116.5° | 94.1° |
| | 2017-2018 | 10246 | 363 | 17.4 | 1.9 | 112.8° | 94.5° |
| 74°-82° S | 2009-2010 | 15144 | 2495 | 17.2 | 21.7 | 112.8° | 101.4° |
| | 2010-2011 | 15052 | 481 | 17.0 | 21.6 | 113.3° | 101.6° |
| | 2011-2012 | 14664 | 672 | 17.0 | 21.5 | 113.5° | 101.7° |
| | 2012-2013 | 14905 | 2440 | 17.1 | 21.5 | 113.3° | 101.6° |
| | 2013-2014 | 14777 | 1409 | 17.2 | 21.7 | 112.7° | 101.3° |
| | 2014-2015 | 14934 | 753 | 17.6 | 22.1 | 111.7° | 100.8° |
| | 2015-2016 | 14876 | 741 | 18.0 | 20.5 | 110.4° | 100.4° |
| | 2016-2017 | 14636 | 2328 | 18.6 | 16.5 | 108.8° | 110.1° |
| | 2017-2018 | 14732 | 1883 | 19.3 | 12.4 | 106.7° | 99.8° |

Ntotal       = Number of samples in latitude band during season (DSS = [-20,+55])

Ncloud      = Number of PMC detections

LTasc       = Average local time for ascending node samples [hr]

LTdesc     = Average local time for descending node samples [hr].  Note that some latitude bands can combine times close to 24 hr and close to 0 hr

SCAasc     = Average scattering angle for ascending node samples

SCAdesc   = Average scattering angle for ascending node samples

**Table 1(c)**

Statistics for S-NPP OMPS NP Northern Hemisphere PMC Seasons, 2012-2018


| Latitude | Season | Ntotal | Ncloud | LTasc | LTdesc | SCAasc | SCAdesc |
|----------|--------|--------|--------|-------|--------|--------|---------|
| 50°-64° N | 2012 | 5148 | 126 | 12.5 | 2.6 | 143.6° | 92.6° |
| | 2013 | 6378 | 119 | 12.5 | 2.6 | 143.5° | 92.6° |
| | 2014 | 6532 | 160 | 12.6 | 2.7 | 143.4° | 92.6° |
| | 2015 | 6415 | 86 | 12.6 | 2.7 | 143.5° | 92.6° |
| | 2016 | 6900 | 124 | 12.5 | 2.6 | 143.4° | 92.6° |
| | 2017 | 7215 | 161 | 12.5 | 2.6 | 143.4° | 92.6° |
| | 2018 | 7238 | 186 | 12.5 | 2.6 | 143.5° | 92.7° |
| 64°-74° N | 2012 | 6472 | 385 | 11.8 | 3.0 | 131.7° | 96.7° |
| | 2013 | 8658 | 796 | 11.8 | 3.1 | 131.8° | 96.9° |
| | 2014 | 8598 | 722 | 11.9 | 3.2 | 131.8° | 97.2° |
| | 2015 | 8476 | 709 | 11.9 | 3.2 | 131.8° | 97.1° |
| | 2016 | 9320 | 201 | 11.8 | 3.1 | 131.8° | 96.9° |
| | 2017 | 9792 | 457 | 11.8 | 3.1 | 131.8° | 96.9° |
| | 2018 | 9837 | 884 | 11.8 | 3.1 | 131.8° | 96.9° |
| 74°-82° N | 2012 | 8695 | 1497 | 9.5 | 4.9 | 119.5° | 106.4° |
| | 2013 | 11552 | 2935 | 9.5 | 4.9 | 119.6° | 106.7° |
| | 2014 | 11244 | 2272 | 9.6 | 5.0 | 119.7° | 107.1° |
| | 2015 | 11142 | 2591 | 9.6 | 4.9 | 119.7° | 106.8° |
| | 2016 | 12363 | 1894 | 9.5 | 4.9 | 119.6° | 106.6° |
| | 2017 | 12985 | 2008 | 9.5 | 4.9 | 119.6° | 106.6° |
| | 2018 | 13024 | 3139 | 9.5 | 4.9 | 119.6° | 106.7° |


Ntotal        = Number of samples in latitude band during season (DSS = [-20,+55])
Ncloud        = Number of PMC detections
LTasc         = Average local time for ascending node samples [hr]
LTdesc        = Average local time for descending node samples [hr]
SCAasc        = Average scattering angle for ascending node samples
SCAdesc       = Average scattering angle for ascending node samples

**Table 1(d)**
Statistics for S-NPP OMPS NP Southern Hemisphere PMC Seasons, 2012-2018

| Latitude | Season | Ntotal | Ncloud | LTasc | LTdesc | SCAasc | SCAdesc |
|----------|--------|--------|--------|-------|--------|--------|---------|
| 50°-64° S | 2012-2013 | 5624 | 37 | 14.3 | – | 136.3° | – |
| | 2013-2014 | 6217 | 15 | 14.4 | – | 135.5° | – |
| | 2014-2015 | 6009 | 23 | 14.5 | – | 135.2° | – |
| | 2015-2016 | 5929 | 11 | 14.4 | – | 135.7° | – |
| | 2016-2017 | 7056 | 28 | 14.3 | – | 135.9° | – |
| | 2017-2018 | 7140 | 45 | 14.3 | – | 135.9° | – |
| 64°-74° S | 2012-2013 | 5652 | 333 | 15.0 | 23.5 | 124.9° | 94.6° |
| | 2013-2014 | 6342 | 135 | 15.1 | 23.6 | 124.4° | 94.6° |
| | 2014-2015 | 6115 | 104 | 15.1 | 23.7 | 124.2° | 94.5° |
| | 2015-2016 | 6024 | 53 | 15.1 | 23.6 | 124.5° | 94.6° |
| | 2016-2017 | 7187 | 251 | 15.0 | 23.6 | 124.7° | 94.6° |
| | 2017-2018 | 7278 | 343 | 15.0 | 23.6 | 124.6° | 94.6° |
| 74°-82° S | 2012-2013 | 9819 | 1781 | 16.9 | 21.5 | 114.3° | 102.3° |
| | 2013-2014 | 11076 | 1022 | 17.0 | 21.5 | 113.9° | 102.0° |
| | 2014-2015 | 10821 | 538 | 17.0 | 21.6 | 113.8° | 102.0° |
| | 2015-2016 | 10631 | 326 | 16.9 | 21.5 | 114.0° | 102.1° |
| | 2016-2017 | 12593 | 1619 | 16.9 | 21.5 | 114.2° | 102.2° |
| | 2017-2018 | 12756 | 1522 | 16.9 | 21.5 | 114.1° | 102.2° |


Ntotal = Number of samples in latitude band during season (DSS = [-20,+55])
Ncloud = Number of PMC detections
LTasc = Average local time for ascending node samples [hr]
LTdesc = Average local time for descending node samples [hr]
SCAasc = Average scattering angle for ascending node samples
SCAdesc = Average scattering angle for ascending node samples

**Table 2(a)**

Regression Fit Results for IWC, Northern Hemisphere, 1979-1997


| Latitude | A($\pm$dA) | $R_{time}$ | B($\pm$dB) | $R_{solar}$ | C | Lag | Trend | Conf | Cycle |
|----------|-----------|-----------|-----------|------------|------|------|-------|------|-------|
| 50-64 N | 0.28($\pm$0.14) | 0.50 | -1.27($\pm$0.87) | -0.44 | 62.1 | 0.5 | **4.8** | 2.3 | -5.5 |
| 64-74 N | 0.47($\pm$0.22) | 0.57 | -6.41($\pm$1.53) | -0.77 | 104.6 | 1.0 | **6.0** | 3.3 | **-20.5** |
| 74-82 N | 0.65($\pm$0.22) | 0.70 | -6.52($\pm$1.38) | -0.82 | 115.2 | 0.5 | **7.2** | 2.8 | **-18.3** |
| 50-82 N | 0.62($\pm$0.21) | 0.70 | -5.89($\pm$1.32) | -0.81 | 108.3 | 0.5 | **7.1** | 2.7 | **-17.3** |


**Table 2(b)**

Regression Fit Results for IWC, Northern Hemisphere, 1998-2018


| Latitude | A($\pm$dA) | $R_{time}$ | B($\pm$dB) | $R_{solar}$ | C | Lag | Trend | Conf | Cycle |
|----------|-----------|-----------|-----------|------------|------|------|-------|------|-------|
| 50-64 N | 0.20($\pm$0.11) | 0.59 | -1.05($\pm$1.09) | -0.45 | 57.9 | 0.5 | **3.4** | 0.9 | -4.5 |
| 64-74 N | 0.42($\pm$0.18) | 0.57 | -0.82($\pm$2.02) | -0.27 | 73.5 | 1.0 | **5.1** | 1.6 | -2.5 |
| 74-82 N | 0.24($\pm$0.18) | 0.44 | -2.21($\pm$1.75) | -0.43 | 98.1 | 0.5 | **2.6** | 1.5 | -5.8 |
| 50-82 N | 0.30($\pm$0.17) | 0.49 | -1.48($\pm$1.66) | -0.36 | 88.8 | 0.5 | **3.3** | 1.5 | -4.1 |


**Table 3(a)**

Regression Fit Results for IWC, Southern Hemisphere, 1979-1997


| Latitude | A($\pm$dA) | $R_{time}$ | B($\pm$dB) | $R_{solar}$ | C | Lag | Trend | Conf | Cycle |
|----------|-----------|-----------|-----------|------------|------|------|-------|------|-------|
| 50-64 S | 0.98($\pm$0.26) | 0.54 | +4.87($\pm$1.92) | +0.19 | 24.9 | 0.5 | **17.3** | 5.1 | **+21.8** |
| 64-74 S | 0.51($\pm$0.23) | 0.59 | -1.06($\pm$1.54) | -0.41 | 70.3 | 0.0 | **7.3** | 4.6 | -3.8 |
| 74-82 S | 0.45($\pm$0.25) | 0.57 | -1.38($\pm$1.65) | -0.44 | 85.3 | 0.0 | **5.4** | 4.5 | -4.2 |
| 50-82 S | 0.53($\pm$0.24) | 0.61 | -0.94($\pm$1.60) | -0.41 | 79.9 | 0.0 | **6.6** | 4.4 | -3.0 |


**Table 3(b)**

Regression Fit Results for IWC, Southern Hemisphere, 1998-2018


| Latitude | A(±dA) | $R_{time}$ | B(±dB) | $R_{solar}$ | C | Lag | Trend | Conf | Cycle |
|----------|--------|------------|--------|-------------|------|-----|-------|------|-------|
| 50-64 S  | -0.08(±0.27) | 0.07 | -2.97(±2.83) | -0.32 | 69.7 | 0.5 | -1.4 | 2.5 | -13.8 |
| 64-74 S  | 0.15(±0.23)  | 0.32 | -3.38(±2.05) | -0.44 | 81.9 | 0.0 | 2.1  | 2.4 | -12.0 |
| 74-82 S  | 0.14(±0.24)  | 0.31 | -4.22(±2.18) | -0.46 | 97.4 | 0.0 | 1.7  | 2.6 | -12.9 |
| 50-82 S  | 0.14(±0.23)  | 0.31 | -3.92(±2.12) | -0.46 | 92.2 | 0.0 | 1.7  | 2.6 | -12.2 |




Multiple regression fit parameters for SBUV merged seasonal average IWC data, using the form

$$IWC = A*(t_{center} - 1979.0) + B*F_{Ly\alpha}(t_{center} - t_{lag}) + C$$

$t_{center}$ = mid-point of PMC season (DSS = [-20,+55]) [years]
$F_{Ly\alpha}$ = Lyman alpha flux averaged over PMC season, scaled by $1 \times 10^{11}$ photons cm$^{-2}$ sec$^{-1}$ nm$^{-1}$
$R_{time}$ = correlation coefficient of secular term
$R_{solar}$ = correlation coefficient of solar term
$t_{lag}$ = phase lag of solar term for fit with smallest $\chi^2$ value [years]
Trend = decadal change in IWC [%]. **Bold** values exceed 95% confidence level.
Conf = amount of decadal change required to exceed 95% confidence level [%]
Cycle = calculated variation in IWC from solar minimum to solar maximum [%], using a
Lyman alpha flux range of $2.6 \times 10^{11}$ photons cm$^{-2}$ sec$^{-1}$ nm$^{-1}$. **Bold** values exceed 95%
significance of regression fit coefficient.