# Peer review of "Extending the SBUV PMC Data Record with OMPS NP"

_Atmospheric Chemistry and Physics, 2018_

## Referee Comment (RC1) · Anonymous Referee #2 · 26 Dec 2018

This paper has extended PMC data using recent OMPS NP observation, to investigate a 40-year PMC variation. Although methods used in the data analysis are not new, their effort is vitally important to reveal a long-term PMC trend, which cannot be captured by a single satellite. The manuscript is mostly well written, but more explanation/discussion would be needed in some points. In particular, it seems that interpretations of the obtained results are missing. Thus, a section "Discussion" would be needed. The reviewer recommends publication after revising the manuscript regarding the comments below.

p.5, I.115-125: The validation is qualitative and insufficient. Can the authors show quantitative validation for the results with some specific definitions? For example, how large are error values for the data points in Figures 3-5? If we can see error values, it

would be easier to make a judgement about "anomalous" or not. BTW, which dataset is anomalous in the case of NH 2016 season at  $64^{\circ}-74^{\circ}N$ ? Do the authors include also some anomalous data in the merging analysis? Please give more careful information about it.

p.9, I.169-175: Can the authors explain what is a scientific reason of the break point? How concrete or confident is the reason? Do we have to consider possibilities for any other break points or no break point?

p.11, I.193-210: Can the authors give more discussion for the obtained results? For example, concerning to (a), what is a scientific reason for "the trends for segment 2 are smaller than those derived in 2015"? Is that a new important finding? In the same way, the authors should have careful reconsiderations for the results from (a) to (d). The reviewer suggests that it should be summarized as a section "Discussion". Otherwise, it would be difficult to understand scientific importance and/or impacts of the main results, i.e., the trend update, compared with DeLand and Thomas (2015).

p.12, I.242-244: Information on data source for OMPS NP is missing.

---

## Referee Comment (RC2) · Anonymous Referee #1 · 3 Jan 2019

Review of "Extending the SBUV PMC Data Record with OMPS NP" by DeLand and Thomas [2018].

This manuscript describes how the new Ozone Mapping and Profiling Suite (OMPS) Nadir Profiler (NP) can observe Polar Mesospheric Clouds (PMC). PMC results from OMPS can furthermore be combined with existing results from a similar suite of observations by the Solar Backscatter Ultraviolet (SBUV) instruments to create and extend a 40-year record of PMC observations. The authors argue that this multi-decadal record can be used for long-term trend studies of the Earth's mesosphere by splitting the record into two segments with a break point imposed in 1998 and analyzing each segment separately.

The OMPS observations are valuable as they complement and could extend the mul-

tidecadal PMC observations by the SBUV instruments, which will be discontinued in 2019-2020. The similarity between the instruments furthermore allows for relatively minor modifications to existing PMC retrieval algorithms currently in place for the SBUV data. However, there are many details lacking in how the authors produce their results and the reviewer requests that the authors satisfactorily address the following concerns before the manuscript can be recommended for publication in ACP. These are divided into Specific Comments and Technical Corrections below.

Specific Comments:

1. Section 1, p. 3, last sentence. The authors need to be more explicit about what they are presenting in this manuscript. To this end, Section 1 needs an additional paragraph at the end motivating what is to come, instead of the final sentence. This paragraph should indicate that in Section 2, the authors compare PMC frequencies and ice water content (IWC) from the two nadir-viewing datasets (SBUV and OMPS). In Section 3, the trend study is done only for IWC (if that is the case) and is split into two different periods, with a break point at 1998. Also, if frequency is not included (i.e. inclusion of IWC values that are zero in the averages shown) then they also need to explain why either here and/or at the beginning of Section 3.

2. Section 2. The native OMPS data are not shown. The reviewer requests an additional figure between Figures 2 and 3 showing a sample day of the total observed OMPS albedo, indicating which data points are PMC and which are not. A figure analogous to DeLand et al. [2003] Figure 2 for the NOAA-9 SBUV/2 data would be appropriate here. In supporting text, a discussion of the detection threshold and systematic uncertainties resulting from the separation of the PMC signal from the bright background signal (perhaps referencing Figure 2 of the present paper) would provide valuable context for future comparisons with more sensitive limb sounders or nadir imagers.

3. Section 2. The comparison between the two very similar nadir viewing instruments

(SBUV and OMPS) does not "validate" (line 114) the results since the instruments, the observational approach, and the retrieval algorithms are quite similar. There are now many observational studies that compare nadir viewing PMC observations, including Bailey et al. [2015], Benze et al. [2018] and Broman et al. [2018]. There are also modeling studies that show variations of cloud frequency and IWC as a function of instrument sensitivity over the diurnal cycle and at a variety of latitudes [e.g. Bardeen et al., 2010; Stevens et al., 2017; Schmidt et al., 2018]. Curiously, none of these studies are discussed or even cited by the authors. Even if the above studies do not represent identical conditions of the SBUV and OMPS shown, the authors could compare their average IWC against average results for similar conditions. This discussion should necessarily include the particle sizes to which OMPS and SBUV are sensitive.

4. P. 4, Lines 95-104. Is the scaling factor applied to the operational SBUV product? If so how would the user go about reproducing the results in the manuscript given that there are additional tests performed to identify PMC (please provide a reference for these tests on line 104). Also, is the SZA dependence due to ice particle scattering on solar scattering angle? If so, the authors should say this and if the solar scattering angle is controlling the variation shown in Figure 2 then that quantity should be on the x-axis rather than SZA. In addition, if this dependence is determined from a single phase function the authors need to state this as well as well as any other assumptions that go into Figure 2.

5. P. 5, Figure 2. Since the quantity relevant to the results reported in this paper is IWC, it would be most instructive to the reader to show that threshold in this figure rather than an albedo threshold. If the IWC threshold is dependent on both the albedo and the solar scattering angle [e.g. DeLand and Thomas, 2015], then this can be done by using a different color and the same line types (solid and broken), referencing new labels on the right-hand axis drawn using the same color.

6. PP. 6-8, Figures 3-5. It is not indicated until the conclusion that the OMPS NP instruments are in sun-synchronous orbits and that information should be indicated in
supporting text for these figures or before. Similarly, the orbital inclination of OMPS NP should be in supporting text for these figures, particularly as it compares to the SBUV suite of instruments. This would help to clarify the latitudinal coverage of each. In Figures 3-5, what are the coincidence criteria in space and time used for the data shown? Furthermore, please indicate explicitly (in the panels and/or the captions) what local times are averaged, whether both nodes SBUV and OMPS are used, the SZA (and/or solar scattering angles) and what days are used to define the season.

7. Section 3. It is curious why the authors compare cloud frequencies and IWC in Section 2 (Figures 3-5) but for the trend results in Section 3 (Figure 6) only IWC is shown. If frequency is not included in the IWC trend results of Figure 6 (i.e. the inclusion of observations for which IWC=0), there needs to be a statement in the text to this end as well as an explanation of this decision. If frequency trend results using the OMPS and/or SBUV data appear elsewhere, then the authors need to cite these studies.

8. P. 8, line 153. Have the authors explored how their IWC trend estimates vary depending on their season duration? Please comment in the text. Similarly, how different are the trends if no normalization adjustment is made (lines 156-166)? Please comment in the text.

9. P. 10, Figure 6, p. 11 top and throughout. Have the authors made any attempt to restrict their IWC trend analysis in local time, as was done by Hervig et al. [2016] and Hervig and Stevens [2014]? If so, how different are there retrieved trends when they do this? If not, they need to state this in the text to help distinguish their results from previous trend studies. Similarly, have the authors made any attempt to reproduce the longitudinally dependent SBUV trends reported by Fiedler et al. [2017]? If not, they need to state this in the text as well.

Technical Corrections:

1. P. 3, Line 73. The local times relevant to this study are those at PMC latitudes in the

NH and SH rather than the Equator-crossing time. Please reword.

2. Figure 2. Additional information is required indicating the data used. This information could be in the figure itself and/or the caption and would include (but not limited to) satellites, seasons, and days used as well as local times (see also #4 and #5 above).

3. P. 5, line 114. Given #3 above, the word "compare" is more accurate than "validate".

4. P. 5, lines 115-116. This is more accurately stated as "7 NH seasons and 6 SH seasons between 2012-2018" (see also #1 above). If they are using the approach of DeLand and Thomas [2015] then they need to state explicitly in the text that IWC is derived assuming a linear relationship with PMC albedo and with fit coefficients derived from general circulation model (GCM) results.

5. P. 9, line 160. If "each instrument" means "each SBUV and OMPS instrument" the authors should say so.

6. P. 9, line 170. There needs to be a more complete explanation about the cause of this change in the late 1990s as context for the reader.

7. P. 11, lines 192-193. Does the statement "those derived in 2015" refer to DeLand and Thomas [2015]? If so, based on Table 4b of that paper a more useful statement for the reader is something like "...although the trends for segment 2 (1998-2018) are smaller than those derived by DeLand and Thomas (2015) over a shorter time period (1998-2013)."

8. P. 11, lines 192, 194, 199, and 204. By "significant" do the authors mean "statistically significant"? If so, the authors should explicitly say this. If not, they need to say what they mean in the text.

9. P. 11, lines 207-208. Please provide the typical number of observations so that the reader has more context for the "10-20 clouds" observed.

10. P. 12, lines 217-218. Is there an explanation for the phase lag in the NH? If not

then a more complete statement is "Both the source of the heimispheric difference in solar activity response and the source of the derived phase lag in the NH are not understood."

11. P. 12, line 228. "above" should be "poleward of".

References

Bailey, S.M. et al.: Comparing nadir and limb observations of polar mesospheric clouds: The effect of the assumed particle size distribution, J. Atm. Sol.-Terr. Phys., 127, 51-65, 2015.

Bardeen, C.G. et al.: Numerical simulations of the three-dimensional distribution of polar mesospheric clouds and comparisons with Cloud Imaging and Particle Size (CIPS) experiment and the Solar Occultation For Ice Experiment (SOFIE) observations, J. Geophys. Res., 115, D10204, doi:10.1029/2009JD012451, 2010.

Benze, S. et al.: Making limb and nadir measurements comparable: A common volume study of PMC brightness observed by Odin OSIRIS and AIM CIPS, J. Atm. Sol.-Terr. Phys., 167, 66-73, 2018.

Broman, L. et al.: Common volume satellite studies of polar mesospheric clouds with Odin/OSIRIS tomography and AIM/CIPS nadir imaging, submitted to Atmos. Chem. Phys. Discuss., 2018.

Schmidt, F. et al.: Local time dependence of polar mesospheric clouds: a model study, Atmos. Chem. Phys., 18, 8893-8908, 2018.

Stevens, M.H. et al.: Periodicities of polar mesospheric clouds inferred from a meteorological analysis and forecast system, J. Geophys. Res. Atmos., 122, 4508-4527, doi:10.1002/2016JD025349, 2017.

---

## Author Comment (AC1) · 15 Mar 2019

Response to reviewers and revised manuscript are attached as a supplement.

Please also note the supplement to this comment:
https://www.atmos-chem-phys-discuss.net/acp-2018-1034/acp-2018-1034-AC1-supplement.pdf

---

## Author Response (AR2)

For final publication, the manuscript should be accepted subject to minor revisions

Review of "Extending the SBUV PMC Data Record with OMPS NP" by DeLand and Thomas [2018].

The reviewer thanks the authors for responding to comments. The reviewer has more comments on these responses before the manuscript can be recommended for publication. Specific comments and technical corrections are below.

**Specific Comments:**

1. The reviewer thanks the authors for including the requested figure (new Figure 3). Please indicate the year, day and UT of the data shown so that the interested reader can reproduce the figure. At the bottom of p. 6 the authors state the spread of the non-PMC albedo residual values equatorward of 60 degrees. If the authors are implying that this is a statistical uncertainty in the residual PMC brightness at all latitudes, then they should say so explicitly in the manuscript. If that is not what they meant then please explain the significance of this spread in the manuscript. Nevertheless, the reviewer has asked for an estimate of the systematic uncertainty at PMC latitudes (i.e. poleward of 60 degrees) and this is still not provided. This is important for quantitative comparison with other satellite datasets (including SBUV) and different from the threshold shown in Figure 2. It is an estimate of the bias introduced when separating the Rayleigh background from a given observation. Please state and justify this systematic uncertainty as a function of PMC latitude.

> The data used in Figure 3 represent all samples for 2018 day 189 (July 8), as indicated in the caption on lines 177-178. As such, there is no single time associated with these data.

> We have clarified that the spread of residual values (line 161) is caused by ozone variability, as discussed further on lines 168-170.

> DeLand et al. (2007) discuss the uncertainty in the derived PMC albedo for a single sample due to numerous components. The largest term in this calculation is the uncertainty of the background fit, which includes geophysical variability due to ozone fluctuations (as discussed above) as well as the accuracy of the polynomial fit. They provide estimated uncertainty values (relative to the PMC albedo) at 273 nm of 32% at SZA = 40°, decreasing to 20% at SZA = 80°. This represents an uncertainty of

~2x10$^{-6}$ sr$^{-1}$ for most latitudes, since SBUV and OMPS PMC detections at lower SZA (and lower latitudes) are typically fainter.

We have conducted separate [unpublished] analysis of variations in the PMC background albedo value at a fixed SZA throughout a typical PMC season.  We note that the latitude corresponding to a selected SZA can vary by 5°-10° during the season, so that the ozone amount (and thus background albedo) will have some inherent natural change during the season as well.  Examination of these data shows a smooth variation through the season at latitudes less than ~70°, which we interpret as typical behavior.  For SZA values that sample the highest latitudes available to SBUV (75°-81°), the background albedo values show an additional increase of ~2-3x10$^{-6}$ sr$^{-1}$ during a period that approximately corresponds to the period of consistent PMC detections at the same latitude.  This may represent a bias due to the presence of fainter PMCs that are "embedded" in the background samples and are not identified.  It is not clear how to further separate such clouds from albedo fluctuations due to ozone variability.  We have added some text at line 183 (following Figure 3) to summarize this analysis.

2. The reviewer thanks the authors for adding text discussing comparison of their results with other datasets and with model results.

a. There is new text following Figure 6 (line 224) addressing this. On line 235 the authors argue that particle sizes observed by SBUV are greater than 35-40 nm based partly on CIPS results. Hervig and Stevens [2014] calculated SBUV particle radii directly and get this answer for northern hemisphere observations. Is there a reason the authors omitted this SBUV particle size analysis? If not, then please reference the study when discussing particle size.

We do reference the Hervig and Stevens (2014) paper on line 232.  We have added discussion of their particle size results.

b. On line 240, do the authors mean Figures 4(b), 5(b), and 6(b) rather than Figures 4(c), 5(c), and 6(c)? Figures 4(b), 5(b), and 6(b) show IWC rather than frequency.

The reviewer is correct.  We have revised the text appropriately.

c. Although the comparison with the Stevens et al. [2017] results indicate that IWC is 20-30% greater than OMPS or SBUV, the frequency comparison is much different with the model results approximately ten times larger than the SBUV observations in NH 2009 (4 LT) (see also Schmidt et al., 2018). Since the authors are showing both IWC and frequency in Figures 4, 5 and 6 this should be included in the discussion. Can the authors explain this model/data difference in frequencies? Hervig and Stevens [2014] suggest a bias in the SBUV background subtraction could play a role (see comment #1 and Figure 3) in a reported frequency difference between SOFIE and SBUV. Please comment in the text.

There are numerous differences between the PMC occurrence frequency results shown by Stevens et al. (2017) (their Figure 6) and the SBUV and OMPS results shown in our Figures 4-6.

1. Stevens et al. (2017) do show peak frequency values at 4 LT, but also show a significant diurnal cycle at most latitudes. SBUV and OMPS measurements are taken on both ascending node (typically 10-13 LT) and descending node (typically 3-5 LT) at most latitudes, and we average those measurements together to get a full season result. This value will typically be lower than a single frequency value at 4 LT.

2. Stevens et al. (2017) present results averaged over July only (DSS = [+10,+40]), whereas our frequency values are averaged over a longer season (DSS = [-20,+55]). DeLand et al. (2003) show the typical seasonal variation for SBUV PMC frequency at different latitudes. This difference in averaging time period can represent a factor of two or more change in occurrence frequency at 75°-80°N.

3. Stevens et al. (2017) determine PMC formation using the "0-D" model developed by Hervig et al. (2009) that considers only bulk thermodynamic properties (T, $H_2O$), and does not consider more localized contributions from gravity wave activity, transport by horizontal winds, and the availability of nuclei for PMC formation. It is not surprising if the SBUV observations, which are affected by these terms, yield a lower PMC occurrence frequency than the idealized approach of the 0-D model.

4. Hervig and Stevens (2014) evaluate the SBUV PMC detection threshold using a fixed albedo threshold of $5 \times 10^{-6}$ sr$^{-1}$ at 252 nm that was originally selected by Thomas et al. (1991). However, the current SBUV algorithm has modified this threshold, and the V3 PMC product that they examine uses the SZA-dependent threshold for 273 nm albedo shown in Figure 2 of our paper (green dot-dash line). At 65°-70°N, that threshold has a value of ~$5 \times 10^{-6}$ sr$^{-1}$ for ascending node measurements (SZA ≈ 45° in this figure), and a value of ~$8.5 \times 10^{-6}$ sr$^{-1}$ for descending node data (SZA = 80°-85°). So their Figure 4(b) includes many points at relatively low scattering angle (high SZA) that are in fact below the threshold used by DeLand and Thomas (2015) for the SBUV V3 product. Our response to comment #1 contains a discussion of possible bias in the SBUV background subtraction. That discussion suggests that we do not see evidence for a significant error in our background fit at 65°-70°N.

5. We have also examined the SBUV data for selected seasons to evaluate why any specific sample was not classified as a PMC, using the flag information provided in each public data file. For two randomly chosen seasons (NOAA-19 NH 2012, NOAA-18 SH 2005-2006), we find that 70-75% of the samples with 252 nm albedo residual values above the threshold shown in Figure 2 are already identified as PMCs.  We plan further analysis of these results to evaluate whether our current detection tests are too conservative.  But we do not see evidence for a factor of two underestimate in PMC identification.

**Technical Corrections:**

1. The reviewer asked for local times of the observations presented rather than equator-crossing local times. This was done in Deland et al. [2007] (their Figure 2) and Hervig and Stevens [2014] (their Figure 5), which are both cited by the authors. The authors indicate that they do not do this because they want to characterize orbit drift. Please explain how the equator crossing local time better describes the relevant orbit drift as opposed to the local time of the observations.

Table 1 now lists the average local time in each latitude band for both ascending node and descending node measurements for each instrument (NOAA-19 SBUV/2 and S-NPP OMPS NP) during each season.  This table is referenced in the text (lines 198-199) and in the caption of Figure 4 (lines 212-213).

The Equator-crossing time is commonly used as a reference for sun-synchronous satellites, since the latitude range of interest for scientific studies (where LT also changes with orbit drift) may vary.  For example, users of stratospheric temperature profiles from NOAA polar orbiters (which also carry SBUV instruments) may be more concerned with LT variations in the tropics than at the high latitudes used for PMC studies.

2. Note on line 114 that the solar zenith angle is the supplement of the scattering angle, not the complement.

The text has been revised as requested.

3. The reviewer requested clarification on "each instrument" (now on p. 14, line 313) and whether that meant "each SBUV and OMPS instrument". Although the authors said they revised the text here, the reviewer does not see the changes. Please clarify.

This correction was inadvertently omitted, and has now been added.

4. In the trend discussion on p. 16 the authors indicate that their definition of "significance" is not as simple as "statistical significance". However, in the abstract on line 18 they indeed use the term "statistically significant" when discussing trends. Please either modify the abstract or use "statistical significance" in the trend discussion.

We have revised the abstract to use the phrase "significant at the 95% confidence level" for consistency with our trend discussion.

5. In the conclusion (now p. 17, line 396) the reviewer requested they use "poleward" instead of "above". The authors stated they revised the text here but the reviewer does not see this revision. Please edit or explain.

This correction was inadvertently omitted, and has now been made.

[revised manuscript text omitted]